# Research on Calculation Method of Radiation Response Eigenvalue of a Single-Chip Active Pixel Sensor

**DOI:** 10.3390/s22134815

**Published:** 2022-06-25

**Authors:** Zhiwei Qin, Shoulong Xu, Hanfeng Dong, Yongchao Han

**Affiliations:** 1School of Resource Environment and Safety Engineering, University of South China, Hengyang 421001, China; qinzw@stu.usc.edu.cn (Z.Q.); hanfengdongusc@163.com (H.D.); 2China Institute of Atomic Energy, Beijing 102413, China; hanyongchao@ciae.ac.cn

**Keywords:** monolithic active pixel sensor, radiation detection, characteristic value, dose rate characterization

## Abstract

In this paper, we present a calculation method for the radiation response eigenvalue based on a monolithic active pixel sensor. By comparing the statistical eigenvalues of different regions of a pixel array in bright and dark environments, the linear relationship between the statistical eigenvalues obtained by different algorithms and the radiation dose rate was studied. Additionally, a dose rate characterization method based on the analysis of the eigenvalues of the MAPS response signal was proposed. The experimental results show that in the dark background environment, the eigenvalues had a good linear response in the region of any gray value in the range of 10–30. In the color images, due to the difference in the background gray values in adjacent color regions, the radiation response signal in dark regions was confused with the image information in bright regions, resulting in the loss of response signal and affecting the analysis results of the radiation response signal. For the low dose rate radiation field, as the radiation response signal was too weak and there was background dark noise, it was necessary to accumulate frame images to obtain a sufficient response signal. For the intense radiation field, the number of response events in a single image was very high, and only two consecutive frames of image data needed to be accumulated to meet the statistical requirements. The binarization method had a good characterization effect for the radiation at a low dose rate, and the binarization processing and the total gray value statistics of the response data at a high dose rate could better characterize the radiation dose rate. The calibration experiment results show that the binarization processing method can meet the requirements of using a MAPS for wide-range detection.

## 1. Introduction

Against the background of the development of advanced nuclear technology, the probability of nuclear accidents and nuclear terrorist attacks is increasing. This poses a great challenge to the accuracy and scope of nuclear radiation detection. At present, scintillation counters or ionization chambers are widely used for nuclear radiation detection as charge particle detectors [1,2,3,4,5], but the equipment and instruments relied on are expensive and have poor environmental applicability, making them difficult to apply to a wide range of complex radiation fields. Therefore, it is very important to find cheap, small and portable instruments and equipment to ensure the detection accuracy and range [6,7,8].

In recent years, with the development of semiconductor technology, due to the low cost, low power consumption, wide response range and easy data acquisition and processing, scholars at home and abroad have begun to use CMOS probes for γ and X-ray radiation detection [9,10,11,12,13], which shows the great application potential of MAPS in the field of radiation environment imaging and nuclear detection. At present, some scholars have studied the noise impact of radiation on maps and the mechanism of charge collection efficiency under a radiation environment [14,15,16,17]. It has been proven that a MAPS has a linear response to radiation fields, and an X-ray transient response mechanism. The latest research also shows the energy transfer process and signal acquisition process of the photoelectric response of a MAPS after being radiated, as well as the different responses of the MAPS to strong radiation and weak radiation [18]. In addition, a MAPS is sensitive to neutron and gamma radiation [19]. However, there is no literature on the relationship between the MAPS output response signal and the radiation field dose rate. This paper aims to further explore the internal relationship between the response signal output by the MAPS after radiation and the radiation level.

For different statistical algorithms, the linear relationship and dispersion degree between the statistical eigenvalue and dose rate of the range interval are different. At the same time, the number of samples of statistical data restricts the detection efficiency and accuracy. This paper presents correlational MAPS radiation response data and a radiation dose rate characterization method of radioactivity level. Through the statistical data of MAPS image samples and the analysis of a statistical eigenvalue algorithm, the linear relationship between different statistical eigenvalues and radiation dose rates within the detection range of the MAPS was studied, and the applicable range and conditions of each method were analyzed. This research can provide data and algorithm support for improving the accuracy of radiation detection methods based on a MAPS.

## 2. Materials and Methods

### 2.1. Experimental Sample

A SONY IMX 222 CMOS single-chip active pixel sensor was selected as the sensor in the experiment. The pixel size was 2.8×2.8 μm and contained $2.43\times 10^{6}$ valid pixels. The sensor was a Bayer array, with an analog voltage of 2.7 V, a digital voltage of 1.2 V, and an interface voltage of 1.8 V. A MAPS was integrated into the sensor board and provided 8-bit data output. The integration time was adjusted between 1/25th of a second and 1/10,000th of a second.

An Ambarella system on a chip (A5S ARM) was used to read MAPS signals and perform digital signal preprocessing. The power supply voltage was 12 V 2 A DC, and data were transmitted through RJ45 network cables. The front end of the MAPS data acquisition system consisted of a MAPS circuit board and a main board, which were connected by an FPC. In the dark image experiment, an opaque plastic shading material was used to cover the sensor surface to isolate visible light interference in the radiation response signal from the MAPS. During the experiment, the aperture, shutter, gain and white balance control systems were manually set and the noise reduction and exposure compensation functions were turned off.

### 2.2. Experimental Conditions

Radiation experiments were carried out in a γ-ray irradiation chamber and a calibration chamber of the China Institute of Atomic Energy. A cylindrical ^60^Co γ-ray radiation source was used in the irradiation chamber, with characteristic energies of 1.17 Mev and 1.33 Mev. The average activity of the radiation source was 130 kCi, the nonuniformity of the radiation field was less than 5%, and the radiation dose rate of the sample was greater than 60 Gy/h. The radiation experiment was carried out at room temperature, and the ambient temperature was 22 °C. The radiation dose rate was obtained by measuring the total dose divided by the irradiation time. The total irradiation dose was recorded by a radiochromic film dosimeter and measured using a spectrophotometer. The γ-ray calibration chamber contained four collimated gamma-ray sources, including one with an activity of 188 mCi and a ^60^Co source and two ^137^Cs radioactive sources with activities of 11.3 and 225 and 2.17 mCi, respectively. Dose rate calibration was performed using Thermo γ-ray detectors ranging from 0.01 μGy/h to 100 mGy/h. During the experiment, the radiation dose rate of γ-rays was controlled by changing the distance between the source and the sensor. The radiation experimental schemes under different conditions are shown in Table 1 and Table 2.

In order to study the influence of the statistical quantity of radiation response data on the measurement results, image data of 1, 2 and 50 frames were selected at a high dose rate level, and image data of 100, 200 and 1000 frames were selected at a low dose rate level for statistical analysis. Table 3 shows the experimental scheme of convergence efficiency of eigenvalues.

### 2.3. Date Processing Methods

In all experiments, the frame rate of the video file was 25 fps, and the Numpy library of the PYTHON programming language was used to read the image data for frame processing. The response pixel count (the number of pixels with gray value change), the total difference in response to pixel gray value (gray value difference of each pixel before and after irradiation), the sum of pixel gray values and the binarized pixel count (the number of pixels greater than the threshold) were used for data processing. The eigenvalue processing expressions were as follows:The eigenvalue of the total difference in image response pixels in frame *i* is
(1)$$S_{i}=\sum_{i=1}^{i=N}V_{iD}-V_{i},$$
where *V_i_**_D_* is the pixel value of the *i*th pixel when the radiation dose rate is *D*, *V_i_* is the background pixel value of the *i*th pixel and *N* is the number of pixels.

The eigenvalue of binarization processing in frame *i* is

(2)$$T_{i}=\{\begin{array}{c} T_{D,Count}+1,T_{i}\geq T_{iD}\\ T_{D,Count},T_{i}<T_{iD} \end{array},$$
where *T_D,Count_* is the cumulative number of binary pixel counts when the dose rate is *D*, *T_i_* is the gray value of the *i*th pixel and *T_iD_* is the set threshold gray value. When the current pixel is greater than or equal to the threshold gray value, the cumulative pixel counts + 1. When the pixel gray value is lower than the threshold gray value, the total pixel count quantity remains unchanged.

The eigenvalue of the response pixel count in frame *i* is

(3)$$T_{i}=\{\begin{array}{c} T_{D,Count}+1,T_{i}=T_{\mathrm{th}}\\ T_{D,Count},T_{i}\neq T_{\mathrm{th}} \end{array},$$
where *T_D,Count_* is the number of response pixels when the dose rate is *D*, *T_i_* is the gray value of the *i*th pixel and *T*_th_ is the set threshold gray value. The gray value of the current pixel is the same as that without radiation, and the number of response pixels remains unchanged. When the gray value of the current pixel is different from that of the one without radiation, the number of pixels counts + 1.

The sum eigenvalue of pixel values of frame *i* is

(4)$$S_{i}=\frac{1}{N}\sum_{i=1}^{i=N}V_{iD},$$
where *V_i_**_D_* is the pixel value of the *i*th pixel when the radiation dose rate is *D* and *N* is the number of pixels.

Early studies showed that, in MAPS data images, there is a lot of dark current and background noise generated by pulse particles, and the range of gray values from 0 to 30 has a great influence on the characterization of image data eigenvalues, Thus, data with a gray value of 15 or lower were removed from all eigenvalues during experimental processing to reduce interference from background noise.

In the process of color image data processing, the global eigenvalue is the statistical eigenvalue based on the full color image data, while the local eigenvalue is the statistical eigenvalue based on a certain area of color image data.

## 3. Results and Analysis

Figure 1 shows the histogram of changes in the frame images in the dose rate range from 51.61 to 479.24 Gy/h when the integration time was 1/100 s and the gain was 6 dB under shading conditions. The figure shows that, as the dose rate increases, the corresponding histogram peaks of each radiation dose rate shift to the right, and the Figure 1a shows a local magnified view with gray values between 250 and 255. It can be seen that, with the increase in dose rate, the number of response pixels is increased. When the camera was irradiated by γ-rays, the typical response events of photon radiation changed from weak to strong with the increase in dose rate. It expresses the transition from a weak photon response event with a central pixel gray value of about 15 to a strong photon response event with a central pixel gray value of about 250. With the increase in dose rate, the peak value shifted to the right, and the gray value began to peak in the range of 250~255. The higher the dose rate, the higher the peak value in the range of 0~25.

For the high dose rate range under dark conditions, the error analysis of the dose rate characterization fitting results of each eigenvalue is shown in Table 4, and the dose rate range is between 51.61 and 479.24 Gy/h. The data in the table show that the fitting results of the four characteristic values selected in group 5# under the dark image integration time of 1/100 s and the gain of 24 dB are better. Comparing the experimental results of group 1#, 2#, and 3#, the fitting results of group 2# are the best when the gain variable is fixed. This indicates that, under the condition of 6 dB gain, the integration time of 1/240 s is the best, and the characterization ability of the eigenvalue dose rate is normally distributed with the integration time. With a fixed integral time variable, the fitting result of group 5# was the best. This indicates that the gain condition of 24 dB has the best effect under the condition of 1/100 s integration time, and the characterization ability of the eigenvalue dose rate is normally distributed with the integration time. The fitting effect of binary pixel count is the best among the four eigenvalues.

Figure 2 shows the nine background regions selected in the color image radiation experiment. The radiation response events in each region are shown in Figure 3. All seven color backgrounds could reflect the radiation response. However, some weak response signals were drowned due to the difference in gray values in different regions. Areas with a high background gray value are not suitable for radiation dose rate characterization; the lower the background gray value, the higher the characterization accuracy of dose rate. In this paper, the radiation experiment was only carried out in a static environment. In order to be suitable for a wider range of radiation detection, an algorithm for detecting the edge of the frame image can be designed in the future, and the areas conducive to dose rate characterization can be preferentially selected by computer image processing technology. Thus, radiation detection in a dynamic environment can be realized.

The error analysis of the fitting results of the global and dark area statistics of the color images is shown in Table 5 and Table 6. The results show that, under bright conditions, the data of the global image are not suitable for fitting and characterization by eigenvalues. This is because there is a large difference in gray values between different regions of the background image, and direct acquisition of data of the global image will cause data in bright areas to drown radiation response signal data. For example, the gray values of g and I are generally around 200 and 140, and response data lower than these gray values will be submerged by the background values of G and I, meaning that the statistical response data cannot correctly represent the radiation field information.

The error analysis results show that the dose rate characterization ability of the global eigenvalues of color images is much lower than that of dark region eigenvalues. According to Table 5, in the global region, only group B1# and group D1# can reflect the radiation field dose rate change well, while Table 6 shows that, in the dark region, only the total gray value eigenvalue of group C2# failed to fit. This is because there are many regions with different gray values in the global image, so the response signal is easily submerged. The gray value of dark areas is generally between 10 and 20, and the gray value of the pixel generating the radiation response event is much higher than this interval, so the response signal is relatively easy to be statistics. This leads to the statistical eigenvalue dose rate characterization efficiency and ability being better in the dark area than in the global area.

Figure 4 shows the low dose rate image data collected from dark images at the integration time of 1/25 s. The dose rate ranged from 0.4466 to 6.1280 mSv/h. It can be seen from the fitting results that the three characteristic values of the response pixel count, the total gray value and the total difference of response pixels had a large degree of dispersion, which cannot correctly represent the change in radiation field dose rate. However, the binarized pixel count showed good linear response characteristics, with the R^2^ value staying above 0.99.

According to the linear response and error analysis results, among the four statistical eigenvalues, the binarized eigenvalue had the highest characterization accuracy for the radiation level of the radiation field. In the dose rate range of 0.4466 to 6.1280 mSv/h and 51.61 to 479.24 Gy/h, all the characteristic values in the global region of the color image were not suitable for the dose rate characterization of radiation detection. However, after selecting the dark region in the global image to binarize the response signal, the eigenvalue could better reflect the dose rate response law. In the dark image irradiation chamber experiment, the binarization eigenvalues and the total gray eigenvalues showed a good linear response, while only the binarization eigenvalues had a good linear response in the dark image calibration chamber experiment. The experimental results show that, for the MAPS device adopted in this paper, after eliminating background noise with a gray value below 15, a binarization eigenvalue with a threshold of 127.5 is suitable for representing the dose rate of the radiation field and can achieve the requirement of dose rate detection for a wide range radiation field.

The different convergence efficiencies of binary eigenvalues are shown in Figure 5, Figure 6 and Figure 7. It can be seen that the fitting results of one frame of image data in the dose rate range of 51.61 to 479.24 Gy/h and 100 frames of image data in the dose rate range of 0.4466 to 6.128 mSv/h have a large degree of dispersion, which is not enough to meet the statistical requirements of eigenvalues. The fitting results of two frames at a high dose rate and 200 frames at a low dose rate were better. After studying the fitting results of different numbers of frames, it was found that the convergence efficiency of arbitrary eigenvalues is better at a high dose rate radiation level than at a low dose rate radiation level. At the high dose radiation level, there were more radiation response pixels, and the fitting result of two frames of image data continuously collected in the irradiation chamber was close to the fitting result of 300 frames of image data, indicating that the data of two frames in the dose rate range from 51.61 to 479.24 Gy/h can meet the requirements of the eigenvalue calculation. At the low dose radiation level, the radiation response pixels were too few, and there may not have even been response pixels in a single frame image. The fitting result of 200 frames of image data in the calibration chamber was close to that of 1000 frames of image data. It shows that the data of 200 frames of images in the dose rate range from 0.4466 to 6.128 mSv/h can meet the requirement of eigenvalue calculation.

## 4. Verification Experiment

A binary eigenvalue calculation method with a threshold value of 127.5 was selected for the calibration experiment. The data under the following conditions were verified: 1/100 s integration time, 24 dB gain condition and high dose rate of 51.61~119.50 Gy/h, and 1/25 s integration time, 6 dB gain condition and low dose rate of 0.5922~6.128 mSv/h. The experimental results are shown in Figure 8, and the steps were as follows:(1)Remove real point (95, 192,983.47) of high dose rate and remove real point (1.762, 6.571) of low dose rate;(2)Calculate fitted function parameters;(3)Check the dose rate at the removal point and compare with measured value.

As shown in Table 7, within the low dose rate range, the dose rate predicted by function (5) is 1.883 mSv/h, and the error with the actual dose rate is only 6.88%.
(5)$$y=A_{1}\cdot e^{\left(x/t_{1}\right)}+y_{0}$$

Since it is difficult to collect radiation response signals at low dose rates, the results are relatively acceptable. At high dose rates, the radiation response signal is saturated and the error between the predicted dose rate and the actual value is less than 3%. The binarization eigenvalues showed good linear response characteristics, with an R^2^ greater than 0.99. It shows that the binary data processing method can meet the requirements of wide-range detection.

## 5. Conclusions

In this paper, the linear relationship between the statistical eigenvalues obtained by different algorithms and the radiation dose rate was studied, and a dose rate characterization method based on the eigenvalue analysis of the MAPS response signal was proposed. The results show that different background regions in the color image could reflect the radiation response. However, due to the difference in regional gray values, some weak response signals were drowned by the background data with high gray values. Areas with a high background gray value are not suitable for radiation dose rate characterization; the lower the background gray value, the better the dose rate characterization. Among the four statistical eigenvalues, the binary eigenvalue had the highest accuracy in characterizing the radioactivity level of the radiation field. It could well reflect the change in radiation field dose rate in the range from 0.4466 to 6.1280 mSv/h and 51.61 to 479.24 Gy/h. All the eigenvalues in the global color image were not suitable for the dose rate characterization of radiation detection, but after binarization processing of the response signal in the dark area, the eigenvalues could better reflect the dose rate response law. In the dark image high dose rate experiment, the binary eigenvalues and total gray eigenvalues had a good linear response, while in the dark image low dose rate experiment, only the binary eigenvalues had a good linear response. For the MAPS devices adopted in this paper, after eliminating background noise with a gray value below 15, binarized eigenvalues with a threshold of 127.5 were determined to be suitable for representing the dose rate of a radiation field, with an R^2^ greater than 0.99, which can meet the requirements of dose rate detection for a wide-range radiation field.

## Figures and Tables

**Figure 1 sensors-22-04815-f001:**
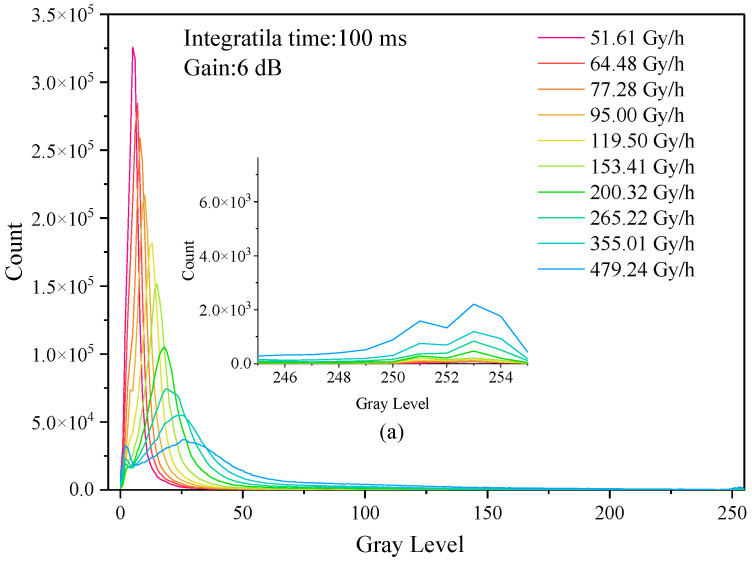
Statistical histogram of dose rate variation. (**a**) belongs to local enlarged figure.

**Figure 2 sensors-22-04815-f002:**
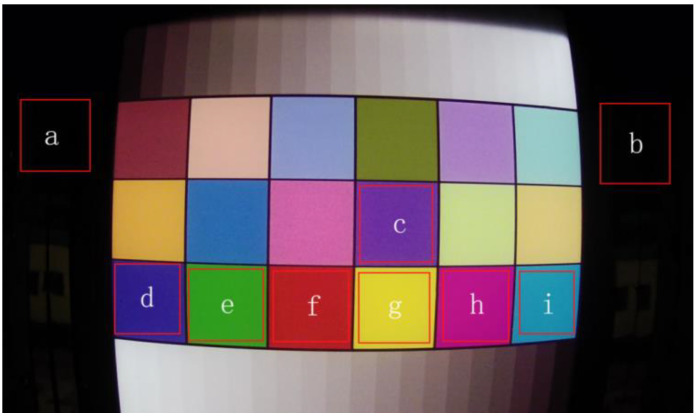
Region division of color image. a and b belong to the dark area, c–i belong to the color area.

**Figure 3 sensors-22-04815-f003:**
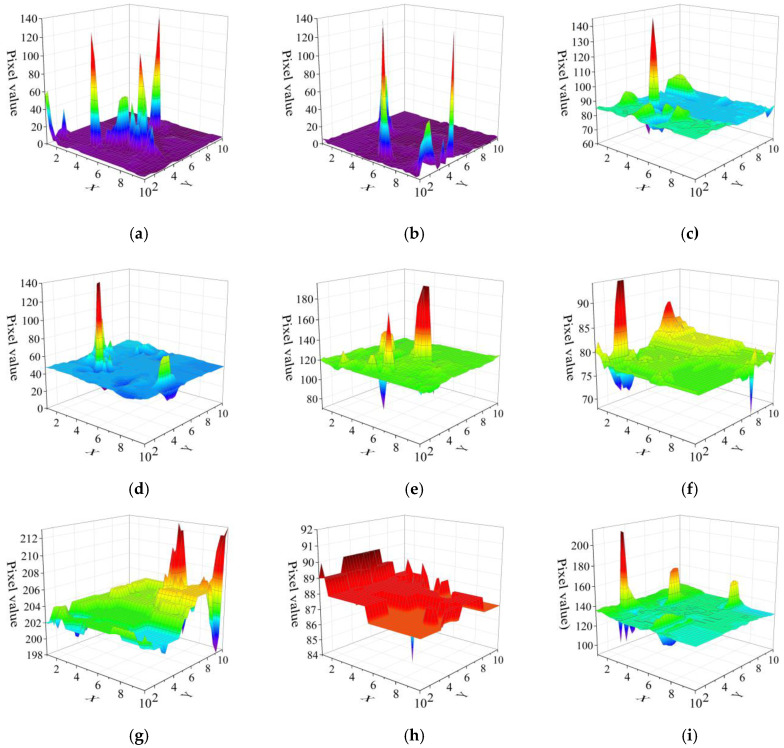
Radiation response events in different color areas. (**a**–**i**) is the response signal diagram of the corresponding area in Figure 2.

**Figure 4 sensors-22-04815-f004:**
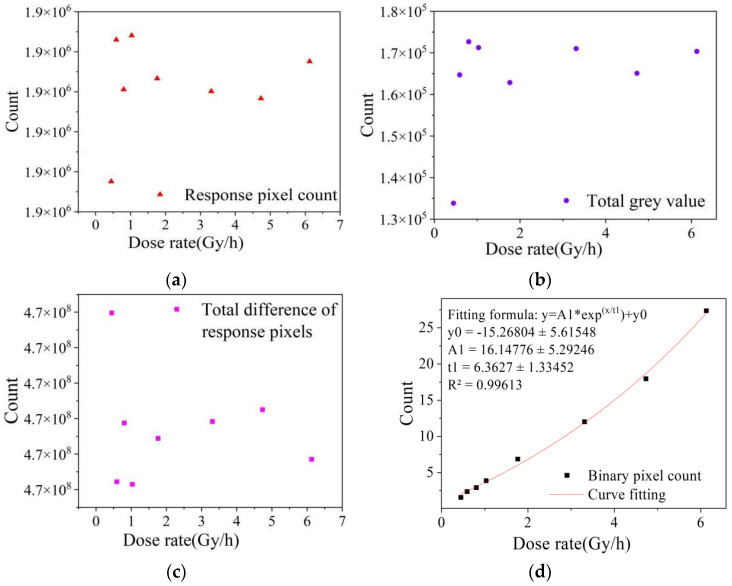
Image fitting data of each eigenvalue frame of calibration chamber. (**a**–**d**) is the fitting data of different eigenvalues in the image.

**Figure 5 sensors-22-04815-f005:**
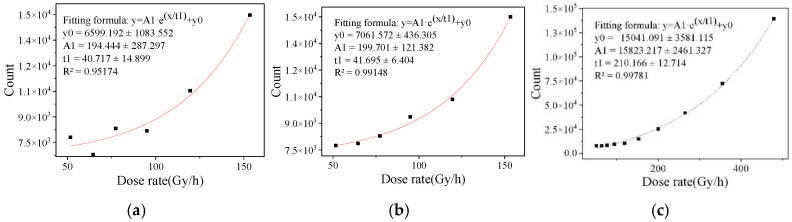
Convergence efficiency of binarized eigenvalue in bright image irradiation chamber. (**a**) 1 frame of image data, (**b**) 2 frames of image data and (**c**) 50 frames of image data.

**Figure 6 sensors-22-04815-f006:**
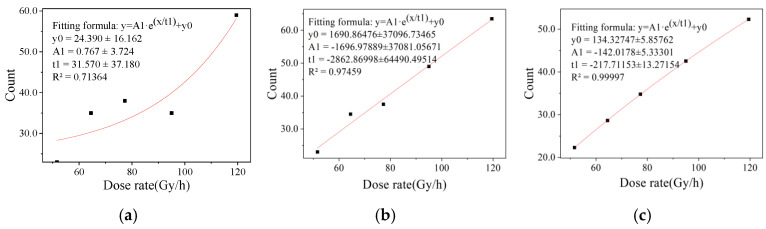
Convergence efficiency of binarized eigenvalue in dark image irradiation chamber. (**a**) 1 frame of image data, (**b**) 2 frames of image data and (**c**) 50 frames of image data.

**Figure 7 sensors-22-04815-f007:**
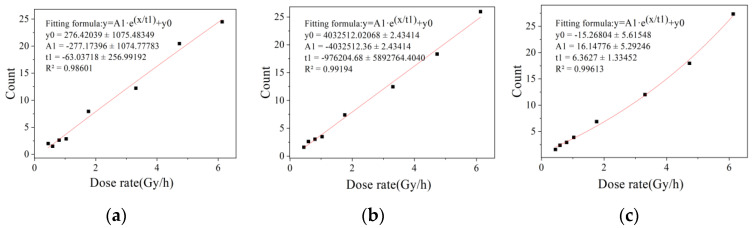
Convergence efficiency of binarized eigenvalue in dark image calibration chamber. (**a**) 100 frame of image data, (**b**) 200 frames of image data and (**c**) 1000 frames of image data.

**Figure 8 sensors-22-04815-f008:**
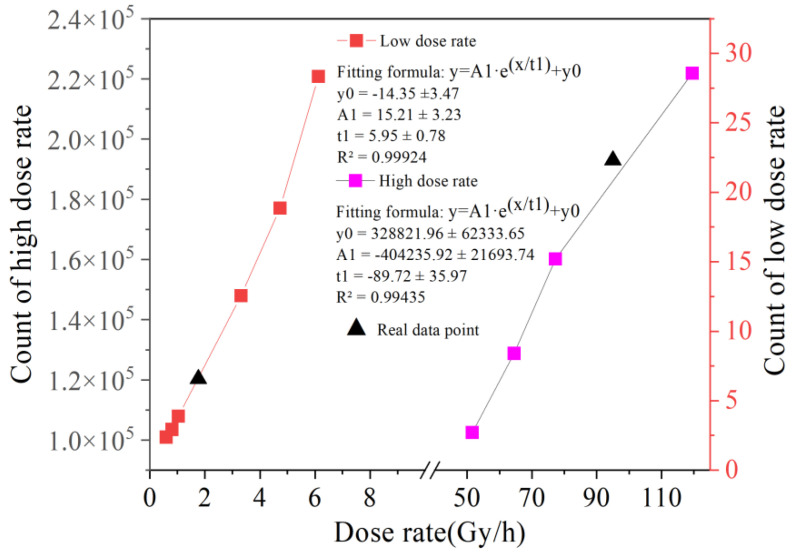
Calibration eigenvalue response curve.

**Table 1 sensors-22-04815-t001:** Radiation experiment scheme of irradiation chamber.

Video Image Collection Type	Radiation Source	Integration Time (s)	Gain (dB)	Dose Rate
Dark	^60^Co	1/100	6	51.61~479.24 Gy/h
	^60^Co	1/240	6	51.61~265.22 Gy/h
	^60^Co	1/480	6	64.48~265.22 Gy/h
	^60^Co	1/100	12	51.61~119.50 Gy/h
	^60^Co	1/100	24	51.61~119.50 Gy/h
	^60^Co	1/100	48	51.61~119.50 Gy/h
Bright	^60^Co	1/8000	6	51.61~119.50 Gy/h
	^60^Co	1/8000	42	51.61~95.00 Gy/h
	^60^Co	1/25	6	51.61~479.24 Gy/h
	^60^Co	1/25	24	51.61~119.50 Gy/h
	^60^Co	1/25	42	51.61~355.01 Gy/h

**Table 2 sensors-22-04815-t002:** Radiation experiment scheme of calibration chamber.

Video Image Collection Type	Radiation Source	Integration Time (s)	Gain (dB)	Dose Rate
dark	^60^Co	1/25	6	446.60 μSv/h~6.13 mSv/h
	^137^Cs	1/25	6	21.92 μSv/h~31.24 mSv/h

**Table 3 sensors-22-04815-t003:** Experimental scheme of convergence efficiency of eigenvalues.

Experimental Conditions	Dose Rate Range	Number of Experimental Frames
Dark irradiation chamber	51.61~479.24 Gy/h	1/2/50
Bright irradiation chamber	51.61~119.50 Gy/h	1/2/50
Dark calibration chamber	0.47~6.13 mSv/h	100/200/1000

**Table 4 sensors-22-04815-t004:** Error analysis of fitting results of characteristic values under different conditions in dark irradiation chamber.

Serial Number	Integration Time (s)	Gain (dB)	Total Gray Value	Response Pixel Count	Binarized Pixel Count	Total Difference in Response to Pixel Gray Value
1#	1/100	6	0.98297	0.97580	0.99792	0.95138
2#	1/240	6	0.99870	0.99693	0.99693	0.99943
3#	1/480	6	0.99360	0.93931	0.93931	0.97832
4#	1/100	12	0.99855	0.99626	0.99919	0.99625
5#	1/100	24	0.99998	0.99992	0.99997	0.99992
6#	1/100	48	0.99996	0.99874	0.98528	0.99822

**Table 5 sensors-22-04815-t005:** Error analysis of global eigenvalue fitting results of color images.

Serial Number	Integration Time (s) and Gain (dB)	Total Grey Value	Response Pixel Count	Binarized Pixel Count	Total Difference in Response to Pixel Gray Value
A1#	1/8000, 6	0.95014	0.98162	0.99853	0.98794
B1#	1/8000, 42	0.99984	0.99994	0.99797	0.99631
C1#	1/25, 6	Fitting failure	0.72182	0.65589	0.63159
D1#	1/25, 24	0.99514	0.99948	0.99124	0.99750
E1#	1/25, 42	Fitting failure	0.84337	Fitting failure	0.52420

**Table 6 sensors-22-04815-t006:** Error analysis of fitting results of feature values in dark area of color image.

Serial Number	Integration Time (s) and Gain (dB)	Total Gray Value	Response Pixel Count	Binarized Pixel Count	Total Difference in Response to Pixel Gray Value
A2#	1/8000, 6	0.99806	0.98918	0.99203	0.99848
B2#	1/8000, 42	0.99987	0.99363	0.99981	0.99981
C2#	1/25, 6	Fitting failure	0.84011	0.99885	0.80715
D2#	1/25, 24	0.98588	0.99541	0.99745	0.99443
E2#	1/25, 42	0.78820	0.80170	0.99776	0.85350

**Table 7 sensors-22-04815-t007:** Calibration error analysis.

Result	High Dose Rate Range	Low Dose Rate Range
Real dose rate	95.000 Gy/h	1.762 mSv/h
Predicted dose rate	97.821 Gy/h	1.883 mSv/h
Error range	2.95%	6.88%

## Data Availability

The data presented in this study are available on request from the corresponding author. The data are not publicly available due to further research.

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
