# Peer review of "Research on Calculation Method of Radiation Response Eigenvalue of a Single-Chip Active Pixel Sensor"

_sensors, 2022, doi:10.3390/s22134815_

Round 1

Reviewer 1 Report

This paper by Qin et al. reported a calculation method of the radiation response eigenvalue for monolithic active pixel sensor output signal. I find this work is suitably conducted and well organized, thus I overall support the publication of this paper at Sensors after a minor revision. The most urgent issue that needs to be addressed is the introduction section. Readers would appreciate if authors could provide an introduction with clear logic transitions and more detailed background of this research. I could follow the introduction well while reading this work. Authors should also check the references and address the issue of self-citations appropriately. After authors make these minor edits, I suggest this work to be accepted for publication. 

Reviewer 2 Report

Paper present method to calculate radiation response eigenvalue basing on monolithic active pixel sensor (MAPS).

Results are interesting, and shows possibility to use MAPS as a radiation monitors. However, before accepting this paper for publication I would like to ask:
what is the meaning of  the fitted curve as presented on Figure 4 (d), Fig 5,6, and 7?
y = A1 * exp(x/t1) + y0 that is connecting binarized eigenalues (y) with dose rate (x). Does it was used as the nearest like function to describe data points? In the section 4 authors verify the method, however it looks like that they obtained just a callibration curve, real check will be done like described below:
a) callibrate the y function with N-1 data points (remove one of the from the callibration)
b) calculate fitted function parameters,
c) check what will be dose rate calculated for Nth point (it should be interpolated one), and compare with measured value.
In this terms we can get goodness of the methos, while in the describtion which is used now it just shows that we can fit defined function to data points.
Can authors show abovementioned check? Otherwise it is not clear if the method is really working well and how well (it will be good to show relative errors for different doses in different conditions - then one can choose optimal condition from that relative dose error and not from goodness of the fit).

I would suggest also to check english language, before publishing.

Round 2

Reviewer 2 Report

I do not have any futher comments or sugggestions after authors make amandements.

I really like the verification experiment results, which shows that the relative error of the method seems to be much lower than 10%.